# The Effect of Forest Video Using Virtual Reality on the Stress Reduction of University Students Focused on C University in Korea

**DOI:** 10.3390/ijerph182312805

**Published:** 2021-12-04

**Authors:** Seong-Hee Jo, Jin-Seok Park, Poung-Sik Yeon

**Affiliations:** 1Forest Welfare Research Center, Korea Forest Welfare Institute, Youngju 36043, Korea; jorent@chungbuk.ac.kr; 2Department of Forest Therapy, Chungbuk National University, Cheongju 28644, Korea; f-welfare@naver.com

**Keywords:** forest video, virtual reality, stress, HRV, EEG

## Abstract

The purpose of this study is to study the effect of forest videos using virtual reality (VR) on the stress of college students. The study subjects were 60 college students who watched two-dimensional (2D) and VR videos, and we compared their control heart rate variability (HRV) and electroencephalogram (EEG). As a result, it was found that the VR group had a positive effect on high frequency (HF), standard deviation of all NN intervals (SDNN), and root-mean-square of successive differences (RMSSD) compared with the control group, and the VR group had a positive effect on HF compared with the 2D group. Second, EEG, a physiological indicator, showed statistical differences in Relative Alpha Power (RA), Relative Beta Power (RB), and Ratio of SMR–Mid Beta to Theta (RSMT) in VR groups in intra-group analysis. Among them, it was investigated that watching forest videos on VR became a state of concentration and immersion due to the increase in RSMT. As a result of the above, it was investigated that forest videos using VR had a positive effect on the physiological stress on college students. Therefore, it is expected that a positive effect will occur if VR is used as an alternative to stress management for college students.

## 1. Introduction

According to the Korea National Statistical Office [1], 7.9% of adolescents aged 20 to 29 said they felt suicidal, which is the highest figure compared to other age groups [2]. In fact, the suicide rate of college students was 44.2% in 2019, 44.2% in 2018, 42.2% in 2017, and 42.9% in 2016 among suicides of adolescents aged 20 to 29 [1]. Suicide research is conducted on all age groups, but it is necessary to pay particular attention to college students. According to statistics on causes of death over the past four years, as shown in Table 1, suicide is the number one cause of death for college students in their 20s [1]. It is a sad reality for young people who will lead the future society, and it has a negative impact on modern society.

Modern college students are greatly influenced by anxiety about their career path [3] and difficulties living in reality [4]. The stressful environment appears as anxiety [5,6], depression [5,6,7], suicide [8], social isolation, etc., and the severity is increasing.

In their 20s, the static influence on suicidal thoughts is influenced by various factors. According to a study by Kim [9], life stress in college students has a positive effect on depression and suicidal thoughts, and various suicide thoughts such as stress with family and friends, suicide with no sense of shame, and defeat [10,11,12,13]. The main reasons for the influence on college students’ suicidal thoughts were various stresses and depression.

Forest landscapes can have a healing effect; they improve health by reducing and recovering human psychological and physiological stress [14,15]. Psycho-evolutionary theory (PET), asserted by Ulrich [14], is a theory that explains how the natural environment reduces response to stress. According to the theory, stress is a physiological reaction to all situations that threaten a healthy life, which causes negative emotions and activation of the sympathetic nervous system. Various natural environments help recover stress, develop an appropriate state of interest, and feel pleasure and tranquility. In the previous situation, negative emotions change to positive emotions, negative emotions soften, and the condition of the sympathetic nervous system decreases. The Attention Restoration Theory (ART), advocated by Kaplan [15], is one of the representative theories that explain the physiological and psychological comfort of the forest environment along with PET. In order for us to continue our daily lives, we must listen to the surroundings and special information. However, attention is gradually reduced due to stimuli generated by external environmental factors or internal psychological factors. In other words, there are external factors that reduce attention in our daily lives, and there is a limit to human attention, so if attention is excessively focused due to external environmental or internal psychological factors for a long time, the capacity of attention is reduced [16]. When shown images of forests, valleys, and trees as well as the sound of wind, water, and birds that can be heard in the forest, the stress condition recovered faster than the urban environment [17]; another study found that viewing natural paintings is more effective in reducing stress [18]. A study by Woo [19] found that indirect forest experience activities and forest image walking, when examining the effect on the mental health of female and humanities high school students, showed improved self-esteem, and humanities high school students had reduced stress and improved psychological well-being. Through many studies, it has been found that the natural environment such as vegetation and water systems becomes an environment that causes stress recovery [20,21,22,23], and the indirect use of forests has a positive effect on the human body [24,25,26].

We are currently living in the era of the Fourth Industrial Revolution. Various modern technologies such as augmented reality (AR), 3D industry, autonomous driving, and virtual reality (VR) are emerging. Virtual reality is a “virtual world that feels real” and is also called an artificial environment or a synthetic environment. It is a space that aims to provide an “as if it is real” experience by taking advantage of the fact that human visual, smell, hearing, and tactile experiences are eventually processed in the brain [27]. The ultimate goal of virtual reality is to make the senses felt by humans in the real world feel the same in the virtual world [28]. One of the most practical technologies among VR is Virtual Reality Therapy (VRT). It provides an opportunity for users to immerse themselves in a realistic but non-realistic treatment environment. Thanks to virtual reality technology, patients can be exposed to situations that are difficult to reproduce, and stimuli that can shift attention from pain can be presented [27]. It is used as various intervention methods using VR. VR is used in various ways such as rehabilitation [29,30,31], sleep [32,33], and pain reduction [34,35,36,37].

The preceding studies above suggest the direction of applying virtual reality technology. In order to obtain the effect of using forests, there is a spatial constraint in that you have to go directly to the mountains or forests. For those who cannot easily go to the mountains or forests, they do not have the effect. Virtual reality, a modern technology that will compensate for these shortcomings, is developing, but previous studies that applied virtual reality to forests are insufficient. Therefore, this study is necessary to study ways to expand opportunities for forest health utility by overcoming the limitations of space to go to the space directly and applying new modern technologies and ways to manage stress, and it is differentiated from existing studies.

The purpose of this study is to investigate the effect of watching forest videos using virtual reality (VR), a modern technology, on the stress reduction of college students and to suggest a stress management plan for college students. The specific objectives of this study are as follows. First, the effect of watching forest videos using VR on heart rate variation (HRV) changes in college students is investigated. Second, the effect of watching forest videos using VR on the electroencephalogram (EEG) change of college students is investigated.

## 2. Materials and Methods

### 2.1. Subjects and Scope of Research

This study conducted a study on college students to study the effect of forest videos using VR on the stress of college students. On 11 November 2020, IRB research approval (CBNU-2011-HR-0176) was obtained from Chungbuk National University’s Institutional Bioethics Committee, and the subjects of the study were college students in Cheongju. The study subjects recruited subjects who had no cardiovascular disease, physical disease, mental illness, and audiovisual abnormalities in watching videos, and the study was conducted with the consent of the study subjects.

Groups were randomly created after the homogeneity of the study subjects was verified. A total of 60 subjects were counted: 20 in the control group without any treatment, 20 in the experimental group watching 2D videos, and 20 in the experimental group watching VR videos. The demographic characteristics are shown in Table 2. All participants conducted the study without dropouts, and 60 subjects were statistically processed. The spatial scope was conducted in the research laboratory of University C in Cheongju-si. It was conducted from 11 November 2020 to 25 December 2020 in the time range, and a total of 4 sessions were conducted in the form of once a week.

### 2.2. Research Progress Procedure

As for the group of this study, a total of three groups were divided into a group that watches VR videos, a group that watches 2D videos, and a control group that does not watch anything. The group watching VR videos watched a forest video taken by GoPro Fusion with GoPro Studio that was converted it into a video that can be viewed 360 degrees. The 2D group viewed their video through a general monitor. The control group had no treatment related to the forest.

All three groups had their heart rate variation (HRV) and electroencephalogram (EEG) measured to verify homogeneity between groups, and a questionnaire on general characteristics was conducted. HRV was tested for 2 min and 30 s, and EEG was measured for 5 min. The group who watched VR and 2D videos watched the video once a week for a total of four times a week for about 5 min per viewing, and the control group proceeded without any treatment. After all experiments were over, the three groups conducted post-examinations (HRV, EEG, and survey), and the procedure for this study is as shown in Figure 1.

### 2.3. Research Tool

#### 2.3.1. 360° Camera

Using the GoPro Fusion (GoPro Inc., San Mateo, CA, USA) camera (Figure 2), 360° images were taken and processed by GoPro Fusion Studio (GoPro Co. Ltd.) Using America, a 360-degree image was produced by stitching two 180-degree images. The direct forest video was filmed with a light walk on the forest-bound deck path for about five minutes, and the filming period was filmed from September to October 2020. The videos used in the study were videos of a total of four locations, and HMD was used and driven through GoPro’s VR player.

#### 2.3.2. HMD VR Device

A Samsung Odyssey+ VR HMD VR device XQ800ZBA-HC2KR (Samsung Inc. Seoul, Korea) was used. It is worn as a headset on the head, and after connecting the HMD device to the computer, the video was played through Window Mixed Reality.

#### 2.3.3. Physiological Examination

Heart rate variability (HRV)

The heart rate variation measurement product used in this study was Ubiomacpa/UBioClip v40 (Biosence Creative Inc., Seoul, Korea) equipment. Using Ubiomacpa (Biosence Creative Co. Ltd., Korea) software, data result values were stored on the computer in real time. Clip-type heart rate variation measuring equipment is equipped with a technology to measure heart rate through light reflection changes in the hemoglobin of fingertip capillaries. The test was conducted in a stable state for about 2 min and 30 s after wearing equipment on the subject’s index finger. To measure changes in autonomic nervous system activity, standard deviation of heart rate and heart rate (standard deviation of all NN intervals, SDNN) and high frequency (HF) and low frequency (LF) of the parasympathetic and sympathetic nervous system were used. Additionally, root-mean-square of successive differences (RMSSD) was used to determine cardiac stability.

Electroencephalogram (EEG)

EEG is a physiological monitoring method that records brain activity from electrodes attached to a person’s scalp. It is non-invasive, and the electrodes are placed along the scalp. It is usually referred to as “EEG”, and it is used to study overall brain-related functions such as sleep research, anesthesia monitoring, and emotional measurement tools. EEG analysis used in this study stored data in real time on a computer through Cygnus (BioBrain Co. Ltd. Korea) software, and the EEG device used was BIOS-ST (BioBrain Inc.) Daejeon, Korea) equipment. Currently, the basic analysis method used is the frequency series power spectrum analysis through Fast Fourier Transformation (FFT). Power spectrum analysis is an analysis method used to determine the pattern of signals according to the degree of frequency change by converting time series signals that change over time into frequency domains [38]. Each equipment, wearing appearance, and international electrode law are as shown in Figure 3.

### 2.4. How to Analyze Data

The statistics of the collected data were analyzed using the statistical program SPSS ver. 18.0.

Frequency analysis was conducted on the general characteristics of the study subjects.

Kruskal–Wallis test and Mann–Whitney’s U-test were performed to verify homogeneity between groups and to find out differences between pre and post.

Wilcoxon code ranking test was performed to compare the differences between the pre- and post-group.

For all analyses, the statistical significance level was set to *p* < 0.05.

## 3. Results

### 3.1. HRV (LF, HF, SDNN, RMSSD)

HRV tests were performed before and after group treatment, and Mann–Whitney’s U-test was performed by analyzing the results between groups. The results are shown in Table 3 and Figure 4. The VR group showed a more positive effect on HF, SDNN, and RMSSD than the control group, and they showed statistical significance (*p* < 0.05). The 2D group showed a greater difference in positive effects on SDNN and RMSSD than the control group (*p* < 0.05), and the VR group showed a more positive effect on HF than the 2D group (*p* < 0.05).

### 3.2. EEG (RA, RB, RSMT)

The descriptive statistics of EEG by group are shown in Table 4. In each electrode indicator, the power value of the Relative Alpha Power Spectrum (RA) decreased in all groups. The power values of the Relative Beta Power Spectrum (RB) and Ratio of SMR–Mid Beta to Theta (RSMT) showed an increase. Among them, the RB and RSMT power values of the VR group increased more than those of the other groups.

The Wilcoxon code ranking test was performed to compare the pre- and post-mortem of VR groups by index within the group, and the results are shown in Table 5 below. In the VR group, the indicators of RB showed statistical significance overall (*p* < 0.05) and partial statistical significance in RA and RSMT (*p* < 0.05). In the VR group, RA tends to decrease, while RB and RSMT tend to increase, as shown in Figure 5, which is mapped. The gauge in Figure 5 means the power value. The red color indicates a higher power value, and the blue color indicates a lower power value. In the case of ‘RA’, the power value is lower than before the experiment, indicating blue. The power values of ‘RB’ and ‘RSMT’ increase to indicate red. The activation of ‘RA’ was lowered, and ‘RB’ and ‘RSMT’ were further activated.

## 4. Discussion

First, heart rate variability was measured first to find out the effect on physiological stress. Comparison between groups showed that the VR group had a more positive significant difference in HF, SDNN, and RMSSD than the control group (*p* < 0.05), indicating that the VR group had a more significant difference in HF increase than the 2D group (*p* < 0.05). As the VR group appears to have a more positive effect on HF, SDNN, and RMSSD than the control group [39], it is considered that the forest video viewed as an image has a physiologically positive effect. This showed similar results to previous studies [40,41,42,43,44,45] in which forest activities reduce stress.

Second, EEG was measured for the second time to find out the effect of physiological stress, and indicators of RA, RB, and RSMT were used. As an intra-group comparison, RA and RB showed statistical significance in overall indicators in the VR group (*p* < 0.05) and partial statistical significance in RSMT (*p* < 0.05). Among them, the increase in beta waves and the increase in RSMT showed that immersion and concentration were concentrated on images physiologically while watching forest videos on VR, similar results to previous studies [46,47], and the decrease in alpha waves decreased when attention was being focused [48].

Forest activities have a positive effect on the psychological aspect of college students [49], as shown in studies in which intervention through forests plays a beneficial mediating role in mental health such as stress and anxiety [50,51]. In addition, a study showed increases in physiological well-being when walking after appreciating bamboo forest paths for 15 min [52], and a study showed that forest rest environments using virtual reality have a physiological positive effect on the human body [53]. Compared to urban areas, it was confirmed through previous studies that forests have a positive effect on the human body, including increased positive moods such as vitality and decreased negative moods [54], and this study confirmed that virtual forests applied with modern technology have an indirect effect on the human body [55].

As a result of the previous study, it was found that watching forest videos on VR provided college students with a state of immersion and concentration, that is, attention, and had a physiological positive effect on stress, and I would like to propose them as a way to reduce stress.

This study has the characteristics of a preliminary study, and as a limitation of this study, the number of samples of the study subjects is small, so it cannot be generalized. There are control variables for the external environment of the study subjects. The period during which this study was conducted can affect the experimental results due to the test period of college students, weather changes, and the increase in COVID-19-confirmed patients, and the period and place of the study should be determined in consideration of external environmental factors. In future studies, directions for increasing reliability through control of the number of samples and the external environment of the study subjects are presented.

## 5. Conclusions

The discussion through the conclusion of this study is as follows.

First, the effect of virtual reality forest videos using VR on the stress of college students is to reduce and have a positive effect on the stress of college students, and I would like to suggest a forest experience using VR as a stress management plan for college students. It was found that the stress of college students is affected in various ways and there is more stress caused by employment. Stress management is necessary because excessive stress increases depression and degrades the quality of college life and daily life.

Second, unlike this study, which moves freely only in sight with a simple 360-degree video, it is expected that producing content that can interact with virtual reality will further feel interest, immersion, and reality and relieve stress.

Third, if a person who can receive the health promotion effectiveness of forests using virtual reality is applied to a different class than only college students, it will have a positive effect on physical disability or elderly care facilities with a lot of indoor life.

Fourth, I would like to propose identifying the mechanisms of beneficial advantages of forests to humans using VR. In this study, vision and hearing were used among human senses. In subsequent studies, it is expected to help verify the utility of forests using sight and hearing.

Fifth, since the effect was verified only with VR images and 2D images, that is, the same forest video, more accurate verification of the effectiveness of the forest video is required. In order to see the effect of forest videos, a more detailed research plan is needed to compare forest videos with other natural videos (sea, sky, etc.) in VR videos to check the effect of forests more accurately, or between groups that directly go through forests and groups that view the same forest as VR videos.

## Figures and Tables

**Figure 1 ijerph-18-12805-f001:**
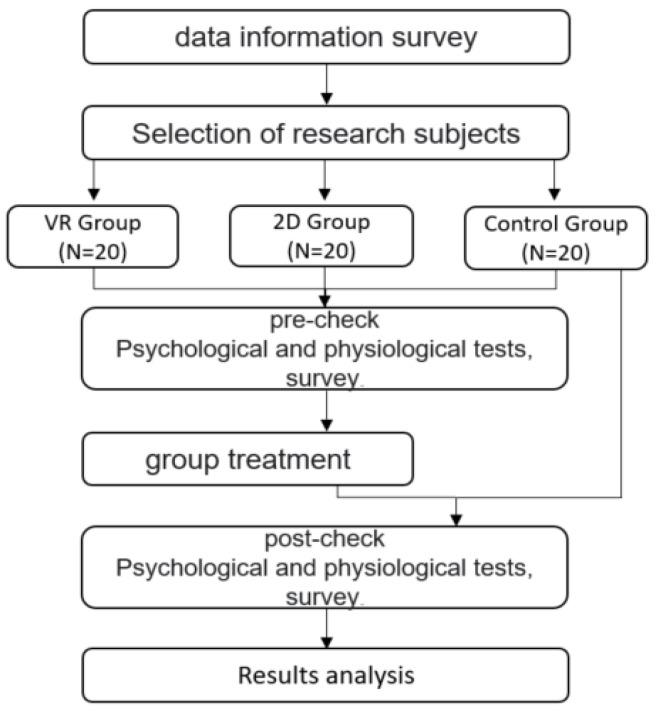
Research process.

**Figure 2 ijerph-18-12805-f002:**
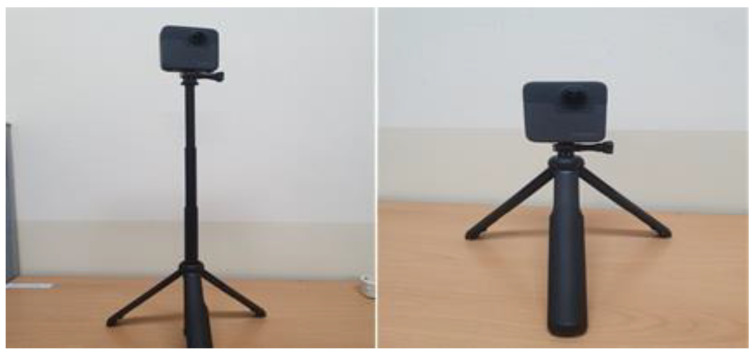
The 360° camera.

**Figure 3 ijerph-18-12805-f003:**
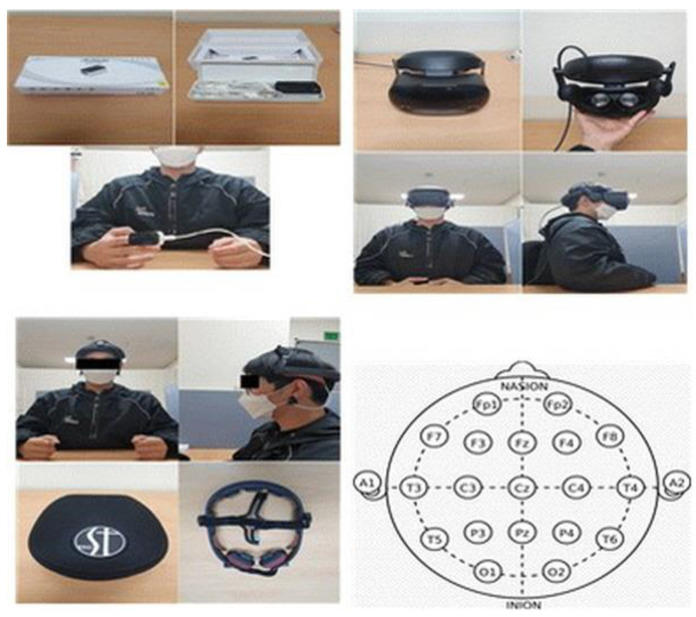
Appearance of wearing experimental equipment and international electrode placement method.

**Figure 4 ijerph-18-12805-f004:**
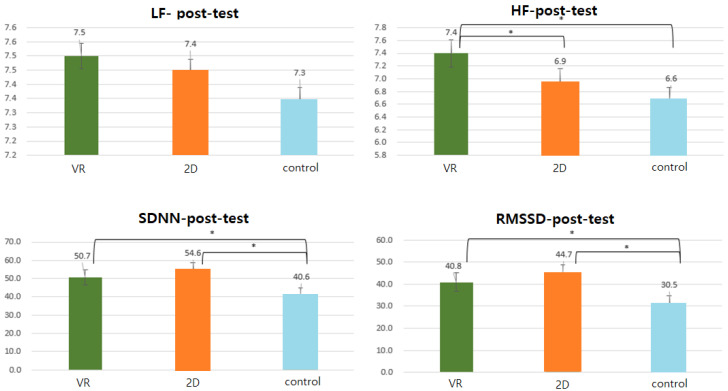
HRV results by group. *: *p* < 0.05.

**Figure 5 ijerph-18-12805-f005:**
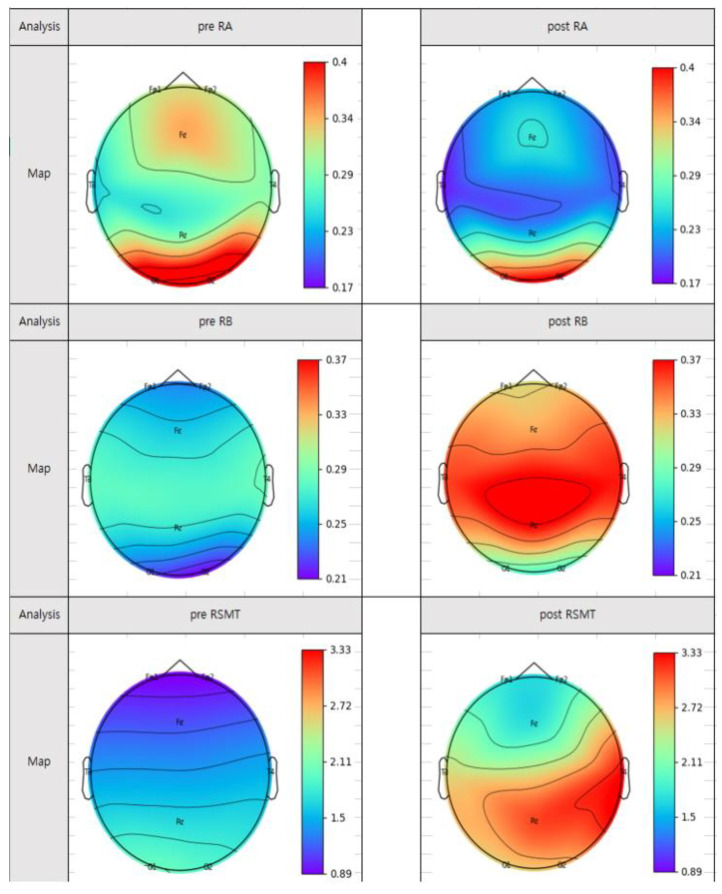
VR group EEG mapping.

**Table 1 ijerph-18-12805-t001:** 2016–2019 Statistics on the cause of death in the 20s (unit: Death rate (per 100,000 people), N).

Year Category	2016		2017		2018		2019	
Suicide	16.4	1097	16.4	1106	17.6	1192	19.2	1306
Traffic accident	5.7	379	5.1	347	4.3	293	3.7	253
Malignant neoplasm	4.2	284	4.0	273	3.9	267	4.2	283
Heart disease	1.5	100	1.5	102	1.5	103	1.4	92

**Table 2 ijerph-18-12805-t002:** Demographic characteristics.

Division	Category	N
Group	VR	20
	2D	20
	Control	20
	All	60
Sex	Man	44
	Woman	16
	All	60
Grade	1	1
	2	13
	3	16
	4	30
	All	60

**Table 3 ijerph-18-12805-t003:** HRV pre–post result.

	VR–Control							
	Pre-Test				Post-Test			
	LF	HF	SDNN	RMSSD	LF	HF	SDNN	RMSSD
U	140.5	191.5	175.5	199.0	173.5	61.0	124.0	122.0
Z	−1.613	−0.231	−0.663	−0.663	−0.718	−3.772	−2.056	−2.110
*p*	0.107	0.818	0.507	0.978	0.473	0.000 *	0.040 *	0.035 *
	2D–Control							
	Pre-test				Post-test			
	LF	HF	SDNN	RMSSD	LF	HF	SDNN	RMSSD
U	176.5	166.5	189.0	183.0	196.0	139.5	119.5	110.0
Z	−0.637	−0.907	−0.298	−0.460	−0.108	−1.641	−2.178	−2.435
*p*	0.524	0.364	0.766	0.646	0.914	0.101	0.029 *	0.15 *
	VR–2D							
	Pre-test				Post-test			
	LF	HF	SDNN	RMSSD	LF	HF	SDNN	RMSSD
U	154.5	164.5	183.5	186.0	179.5	125.0	190.5	190.5
Z	−1.233	−0.962	−0.446	−0.379	−0.556	−2.036	−0.257	−0.257
*p*	0.218	0.336	0.655	0.705	0.578	0.042 *	0.797	0.797

*: *p* < 0.05; LF: low frequency; HF: high frequency; SDNN: standard deviation of all NN intervals; RMSSD: root mean square of successive differences.

**Table 4 ijerph-18-12805-t004:** EEG descriptive statistics.

Category	Group	Pre-Test								Post-Test							
		Fp1	Fp2	T3	T4	O1	O_2_	Fz	Pz	Fp1	Fp2	T3	T4	O1	O2	Fz	Pz
RA	VR	0.317	0.319	0.240	0.286	0.452	0.478	0.345	0.284	0.226	0.223	0.176	0.192	0.386	0.403	0.254	0.227
	2D	0.292	0.282	0.221	0.254	0.388	0.429	0.339	0.268	0.229	0.224	0.162	0.182	0.353	0.373	0.280	0.190
	C	0.241	0.236	0.212	0.234	0.362	0.378	0.271	0.315	0.228	0.228	0.176	0.199	0.345	0.368	0.251	0.233
RB	VR	0.244	0.242	0.283	0.288	0.228	0.212	0.259	0.269	0.322	0.328	0.359	0.368	0.288	0.281	0.336	0.373
	2D	0.282	0.284	0.313	0.321	0.268	0.269	0.270	0.305	0.320	0.320	0.341	0.362	0.301	0.290	0.309	0.387
	C	0.298	0.316	0.327	0.330	0.292	0.285	0.333	0.318	0.306	0.310	0.341	0.343	0.290	0.272	0.313	0.339
RSMT	VR	0.891	0.894	1.403	1.492	2.011	1.901	1.143	1.690	1.733	1.787	2.542	3.333	2.604	2.700	1.742	3.110
	2D	1.346	1.538	1.495	1.886	2.368	2.700	2.006	2.360	1.465	1.530	2.067	2.357	2.651	2.694	1.493	3.069
	C	1.868	1.937	2.182	3.020	2.631	2.838	2.194	2.380	1.789	1.730	2.265	2.536	2.553	2.491	1.951	2.734

RA: Relative Alpha Power Spectrum; RB: Relative Beta Power Spectrum; RSMT: Ratio of SMR–Mid Beta to Theta.

**Table 5 ijerph-18-12805-t005:** VR group post-test EEG analysis.

Category		Fp1	Fp2	T3	T4	O1	O2	Fz	Pz
RA	Z	−3.621a	−3.696a	−3.323a	−3.808a	−2.389a	−2.688a	−3.211a	−1.792a
	*p*	0.000 *	0.000*	0.001 *	0.000 *	0.017 *	0.007 *	0.001 *	0.073
RB	Z	−3.472a	−3.696a	−3.285a	−3.621a	−3.248a	−3.472a	−2.875a	−2.875a
	*p*	0.001 *	0.000 *	0.001 *	0.000 *	0.001 *	0.001 *	0.004 *	0.004 *
RSMT	Z	−3.024a	−3.099a	−2.987a	−2.875a	−1.829a	−2.128a	−2.539a	−2.501a
	*p*	0.002 *	0.002 *	0.003 *	0.004 *	0.067	0.033 *	0.011 *	0.012 *

*: *p* < 0.05; RA: Relative Alpha Power Spectrum; RB: Relative Beta Power Spectrum; RSMT: Ratio of SMR–Mid Beta to Theta; a: Based on the positive number.

## Data Availability

The data presented in this study are available on request from the corresponding author. The data are not publicly available due to privacy.

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
