# Peer review of "The Effect of Forest Video Using Virtual Reality on the Stress Reduction of University Students Focused on C University in Korea"

_ijerph, 2021, doi:10.3390/ijerph182312805_

Round 1
Reviewer 1 Report
The paper entitled as “The Effect of Forest Video Using VR (Virtual Reality) on the Stress Reduction of University Student” simply analyzed effectiveness of VR from some viewpoints. Firstly, I would suggest to improve how to use abbreviation such as HRV, EEG, HF, etc. so it is very difficult to read smoothly. Outcomes of this study would be acceptable in this journal.
General comments: Again, the authors should improve how to use “technical terms” with abbreviation.
Author Response
Dear Editor and reviewer,
We would like to express our sincere gratitude for your kind consideration and comments on our manuscript. According to reviewers’ comments and suggestions, we revised the manuscript as follows:
(We marked the revision to the reviewer's comment in purple)
I modified it to know what it is by describing the abbreviations in the abstract and text. (line 3-18, 131,165 etc.).
Once again, thank you very much for the time spent and the interest shown in this work, and the positive evaluations you have given of it.
Reviewer 2 Report
The article is interesting and the proposed proposal, the relationship between stress reduction and health improvement with the beneficial effects of forest landscapes is well justified. The inclusion of augmented reality work is innovative.
Title, abstract and keywords are tailored to the article content. It is advisable to explain acronyms the first time they appear in the text.
After reading the theoretical framework, two questions appear:
- Of the adolescents between 20-29 years old who considered themselves suicidal, is it possible to know what percentage refers to university students?
- Are there studies that compare the reduction of stress after the use of augmented reality in people who live in wooded environments and those who live far from them?
The bibliographic review developed is adequate and the use of relevant and updated references is observed.
The sample size of the investigation is small. It could have been clarified in the Method section plus the sample and not at the beginning of the Results. What university studies did the participating sample take?
Good specification of the instruments used and includes how to proceed with the data analysis. The authors indicate ethical considerations and required authorizations. Updated research, developed at the end of 2020.
Delete the following text (lines 132-139):
“Research manuscripts reporting large datasets that are deposited in a publicly available database should specify where the data have been deposited and provide the relevant accession numbers. If the accession numbers have not yet been obtained at the time of submission, please state that they will be provided during review. They must be provided prior to publication.
Interventionary studies involving animals or humans, and other studies that require ethical approval, must list the authority that provided approval and the corresponding ethical approval code”.
In the Results, the distribution of the sample by research groups is explained again (repetitive). Table 2 is unnecessary, such information could be included in the characteristics of the sample (Method). The results in Table 3 and Figure 4 could be explained more extensively, including related investigations. The visualization of Figure 4 should be improved and a brief explanation of Table 4 should be made. The results of Table 5 and their reflection in Figure 5 are very interesting.
The Discussion begins with the exposition of the main results obtained, they are not related to other investigations. They include a proposal for future research and research limitations appear. As a limitation, the small sample size could also have been incorporated.
Finally, they provide a wide variety of bibliographic references, a high percentage of which are from the last five years.
Author Response
Dear Editor and reviewer,
We would like to express our sincere gratitude for your kind consideration and comments on our manuscript. According to reviewers’ comments and suggestions, we revised the manuscript as follows:
(We marked the revision to the reviewer's comment in Blue, The common comments of other reviewers are first marked in the color of the reviewer who gave the opinion.)
1. we modified it to know what it is by describing the abbreviations in the abstract (line 13-18, 131,165 etc).
2. There are no statistics on the percentage of college students among adolescents aged 20 to 29 who thought of committing suicide. Instead, among youth suicides aged 20 to 29, there are statistics showing 44.2% of college students' suicides in 2019, 44.2% in 2018, 42.2% in 2017, and 42.9% in 2016, so it was further described in the introduction. (line 32-34)
3. No research has been found comparing stress reduction after using augmented reality for people living far away from those living in forested environments. However, there is a study comparing stress in forests and cities in virtual reality. Fatigue increased and self-esteem decreased in the city. In the forest environment, the level of vitality increased and the level of negative emotions decreased. (Yua,; C, Lee,; H, Luo,; X, The effect of virtual reality forest and urban environments on physiological and psychological responses, Urban Forestry & Urban Greening,2018, 35(2018), 106-114)
4. The students who participated in the study were students from various majors such as economics, humanities, and engineering.
5. I'm sorry. There was a mistake in editing. I deleted the contents. (line 132-139 deleted)
6. Table 2 has been moved from Results to Methods and the content has been corrected. (line 117,122)
7. We improved the visualization of Figure 4 and added additional explanations in Table 4.(figure 4, line 205-212)
8. We added content that cannot be generalized due to the small number of samples of the study subjects and limitations of the study.(line 255-263)
Once again, thank you very much for the time spent and the interest shown in this work, and the positive evaluations you have given of it.
Reviewer 3 Report
I appreciate this documented article and thank you for the opportunity to evaluate it.
In my opinion, VR is a very useful tool for teaching. Having in mind the vertigo feeling and other adverse effects after using VR technology for a long time I have some doubts that VR can be used for relaxation. I would rather recommend a sport for students than VR relaxation. Anyway, the figures show that VR had a positive effect on students' psychology. But the authors have to prove that the sample is representative. This article presents rather preliminary research. None of the conclusions can be extrapolated over the entire population if the sample is not representative.
The authors should take into account also next issues:
Title: You may keep VR or Virtual reality, but not both. You may specify that it is a case study on a specific institution in Korea.
In the abstract, there are a lot of unexplained acronyms HF, SDNN, and RMSSD.
Is the sample representative: 60 college students?
What do you mean by this paragraph: Research manuscripts reporting large datasets that are deposited in a publicly available database should specify where the data have been deposited and provide the relevant accession numbers. If the accession numbers have not yet been obtained at the time of submission, please state that they will be provided during review. They must be provided prior to publication. ? Who are you talking with?
The authors can change the syntagm: "pre- and post-mortem".
Succes with your paper!
Author Response
Dear Editor and reviewer,
We would like to express our sincere gratitude for your kind consideration and comments on our manuscript. According to reviewers’ comments and suggestions, we revised the manuscript as follows:
(We marked the revision to the reviewer's comment in green, The common comments of other reviewers are first marked in the color of the reviewer who gave the opinion.)
1. We added content that cannot be generalized due to the small number of samples of the study subjects (line 255-263).
2. In the title of the thesis, the subtitle 'Focus on C universities in Korea' was added (line 3,4).
3. we modified it to know what it is by describing the abbreviations in the abstract (line 13-18, 131,165 etc.).
4. This study has the characteristics of a preliminary study and was set as the number of samples to see its effectiveness. The title of the paper and the limitations of the study were set and the contents were displayed.(line 3-4, 255-256)
5. I'm sorry. There was a mistake in editing. I deleted the contents. (line 132-139 deleted)
6. we revised the words for the pre-test and post-test(table 3-5).
Once again, thank you very much for the time spent and the interest shown in this work, and the positive evaluations you have given of it.
Round 2
Reviewer 3 Report
Congratulation for changes
In the title choose Virtual Reality or VR, but not both.
You may add more explanation for Fig. 5.
Please add a paraph to say that VR is only one method to reduce stress for students. It is proved that sports, breathing, positive socialization, alimentation, culture, etc would have also a great impact on students' health - maybe greater than VR!!!.
Author Response
We would like to express our sincere gratitude for your kind consideration and comments on our manuscript. According to reviewers’ comments and suggestions, we revised the manuscript as follows:
(We marked the revision to the reviewer's comment in green)
1. ‘Virtual reality’ was selected from the study title.
2. we gave an additional explanation of Figure 5.(line 222-227)
Once again, thank you very much for the time spent and the interest shown in this work, and the positive evaluations you have given of it.
We would like to express our sincere gratitude for your kind consideration and acceptance of our manuscript.